# Effects of fecal microbiota transfer on blood pressure in animal models: A systematic review and meta-analysis

Lingyu Lin[1], Shurong Xu[2], Meiling Cai[1], Sailan Li[1], Yaqin Chen[2], Liangwan Chen[1,3]*, Yanjuan Lin[1,4]*

1 Department of Cardiovascular Surgery, Fujian Medical University Union Hospital, Fuzhou, Fujian Province, China, 2 School of Nursing, Fujian Medical University, Fuzhou, Fujian Province, China, 3 Key Laboratory of Cardio-Thoracic Surgery (Fujian Medical University), Fujian Province University, Fuzhou, Fujian Province, China, 4 Department of Nursing, Fujian Medical University Union Hospital, Fuzhou, Fujian Province, China

* fjxhyjl@163.com (YL); fjxhlwc@163.com (LC)

**Data Availability Statement:** All relevant data are within the manuscript and its Supporting information files.

## Abstract

### Background

Numerous recent studies have found a strong correlation between intestinal flora and the occurrence of hypertension. However, it remains unclear whether fecal microbiota transfer might affect the blood pressure of the host. This study aimed to quantify both associations.

### Methods

An electronic search was conducted in PubMed, EMBASE, Cochrane Library, Web of Science, China National Knowledge Infrastructure (CNKI), WanFang database, Weipu, Embase, and SinoMed to retrieve relevant studies. The final search was completed on August 22, 2022. Two authors independently applied the inclusion criteria, extracted data, and assessed the risk of bias assessment. All data were analyzed using RevMan 5.4.

### Results

A total of 5 articles were selected for final inclusion. All studies were assessed as having a high risk of bias according to the SYRCLE risk of bias tool. The meta-analysis results showed that transplantation of fecal bacteria from the hypertensive model can significantly improve the host's systolic pressure (MD = 18.37, 95%CI: 9.74~26.99, $P$<0.001), and diastolic pressure (MD = 17.65, 95%CI: 12.37~22.93, $P$<0.001). Subgroup analyses revealed that the increase in systolic pressure in the hypertension model subgroup (MD = 29.56, 95% CI = 23.55–35.58, $P$<0.001) was more pronounced than that in the normotensive model subgroup (MD = 12.48, 95%CI = 3.51–21.45, $P$<0.001).

### Conclusion

This meta-analysis suggests a relationship between gut microbiota dysbiosis and increased blood pressure, where transplantation of fecal bacteria from the hypertensive

**Funding:** This work was supported by Fujian Provincial Department of Finance (NO.2021XH019); Fujian Provincial Center for Cardiovascular Medicine Construction Project (NO.2021-76) and Key Laboratory of Cardio-Thoracic Surgery (Fujian Medical University), Fujian Province University.

**Competing interests:** The authors have declared that no competing interests exist.

model can cause a significant increase in systolic pressure and diastolic pressure in animal models.

## 1 Introduction

Hypertension is a common chronic non-communicable disease and a major risk factor for cardiovascular disease, chronic kidney disease, cerebrovascular disease, and coronary heart disease [1]. Epidemiological studies have revealed that the number of people suffering from hypertension will reach 1.5 billion by 2025 [2], placing a huge burden on society from a public health perspective. Hypertension, which is influenced by genetics and environmental factors [3], has been a hot topic in research on its pathogenesis and treatment for years. Recent research suggests a strong correlation between the development of hypertension and the gut flora and its metabolites. Previous research has indicated that changes in the gut microbiome, such as decreased diversity and abundance of gut microbiota, decreased butyrate-producing, and an increase in lactate-producing bacteria, could lead to the development of hypertension [4–6]. Thus, gut flora has become a promising target for the therapy of hypertension, attracting a lot of attention.

Fecal microbiota transplantation (FMT) is an emerging treatment that transfers functional bacteria from normal feces to the gastrointestinal tract, reconstructs new gut flora, and restores host function [7]. Recently, FMT has been used to treat diseases such as Clostridium difficile infection, irritable bowel syndrome, and ulcerative colitis [8–10]. Increasing numbers of clinical trials and animal experiments are also supporting the possibility of utilizing FMT to treat metabolic syndrome [11, 12], including diabetes, obesity, and hypertension. However, to date, the relationship between gut microbiota and hypertension has not been extensively studied. Many animal experiments provide strong support for verifying the causal relationship between the two. Experimental studies in animal models suggest that FMT can significantly reduce blood pressure (BP) with hypertension [13], while others have reported that FMT does not affect BP [14]. Furthermore, dysbiosis of gut microbiota caused by fecal microbiota transfer can directly affect BP (transplanting fecal bacteria from hypertensive donors to normotensive recipients can result in elevated BP) [6, 13, 14]. However, Piotr Konopelski et al. [15] pointed out that FMT does not affect BP or cause long-term changes in the composition of gut bacteria, and host genotype and/or phenotypes may have a greater effect on gut bacteria than gut bacteria have on the host. The reasons underlying these inconsistent results remain unclear and may be related to the metabolites of intestinal flora.

To date, there have been no previous attempts to quantify the impact of FMT-induced changes in intestinal flora on BP in animal models. A systematic review of preclinical research has been proven to be useful for translational medicine and making precise medical care decisions [16]. Additionally, a systematic review of preclinical evidence could inform design, contribute to the success of future clinical studies, demonstrate the necessity for further research, and reduce unnecessary study replication [17]. Therefore, the purpose of this study is to synthesize the effect of fecal bacteria transfer on BP in animal models and provide clues for future research.

## 2 Results

### 2.1 Literature search

The initial search yielded 5735 articles from databases, with an additional two articles retrieved through hand searching. After removing duplicate articles using EndNote software, 4582 articles remained for review. After reviewing the titles and abstracts, 4535 articles that were unrelated to the aim of the study and did not meet the study criteria were excluded. Another 42 articles were

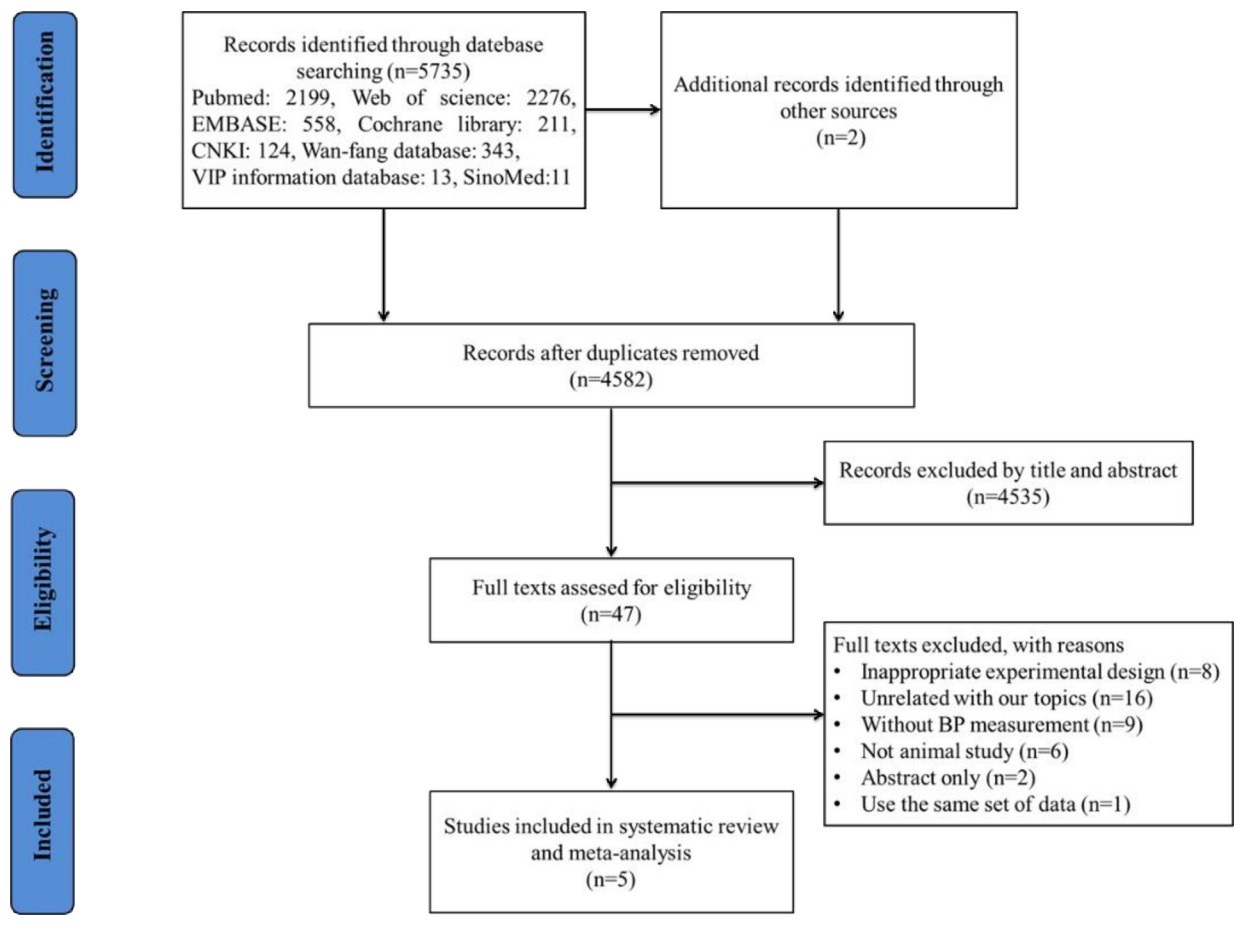

**Fig 1. PRISMA flowchart of included studies.**

further excluded after reading the full text, leaving five articles (involving a total of eight trials) [6, 13–15, 18] to be included in the current study, all of which were in English (Fig 1).

## 2.2 Study characteristics

The characteristics of the included studies are shown in Table 1. The studies were conducted in the United States [14, 18], Spain [13], Poland [15], and China [6], and were published between 2015 and 2021. Of the enrolled studies, four involved rat models [13, 14, 18], and one involved a mouse model [6]. The sample size for each study ranged from 8 to 15, with a total of 109. In all but one article [6], the majority of the included studies administered antibiotics to the participants before FMT. Of the five studies, three used frozen feces [13, 14, 18], and two used fresh feces [6, 15]. The route of FMT included oral gavage, oral inoculation, intracolonic administration, and oral administration.

Of the eight trials included, three were performed in Wistar-Kyoto (WKY) rats [13–15], and the feces were obtained from WKY, spontaneously hypertensive stroke-prone rats (SHRSP), or spontaneously hypertensive rats (SHR). Two trials were performed in SHR [13, 14], and the feces were obtained from WKY, SHRSP, or SHR. One trial was performed in C57BL/6L mice [6], and the feces were obtained from patients with hypertension or

**Table 1. Characteristics of animal experiments.**

| Author | Year | Country | Animals (species, sex, age) | Antibiotic pretreatment | N (T) | N (C) | Intervention | Control | Route of delivery | Fresh/ Frozen | Times of infusion | Dose of infusion | Outcome measurement | Measurements time |
|---|---|---|---|---|---|---|---|---|---|---|---|---|---|---|
| Sareema Adnan 1 | 2017 | The United States | WKY rats, NA, 3.5 week | 10-day antibiotic cocktail | 7 | 7 | supernatant from SHRSP cecal and colon suspension | supernatant from WKY cecal and colon suspension | gavage | frozen | 4 consecutive days and weekly thereafter (11-week) | 750 µl of the cecal and colonic supernatant /rat/time | SBP(tail-cuff blood pressure system) | every 7–10 days for 7 weeks, and a final measurement taken at 16.5 weeks old |
| Sareema Adnan 2 | 2017 | The United States | SHR rats, NA, 3.5 week | 10-day antibiotic cocktail | 6 | 6 | supernatant from SHRSP cecal and colon suspension | supernatant from WKY cecal and colon suspension | gavage | frozen | 4 consecutive days and weekly thereafter (11-week) | 750 µl of the cecal and colonic supernatant /rat/time | SBP (tail-cuff blood pressure system) | every 7–10 days for 7 weeks, and a final measurement taken at 16.5 weeks old |
| Marta Toral 1 | 2019 | Spain | WKY rats, Male, 25 week | 5-day antibiotic treatment | 6 | 6 | supernatant from SHR fecal contents | supernatant from WKY fecal contents | oral gavage | frozen | 3 consecutive days, and once every 3 days for a total extension of 4 weeks | 1ml fecal contents/ rat/time | SBP, DBP, HR (left carotid artery direct register) | at the end of the experimental period |
| Marta Toral 2 | 2019 | Spain | SHR rats, Male, 25 week | 5-day antibiotic treatment | 6 | 6 | supernatant from SHR fecal contents | supernatant from WKY fecal contents | oral gavage | frozen | 3 consecutive days, and once every 3 days for a total extension of 4 weeks | 1ml fecal contents/ rat/time | SBP, DBP, HR (left carotid artery direct register) | at the end of the experimental period |
| Jing Li | 2017 | China | C57BL/6L mice, Male, 8–10 week | NA | 10 | 5 | fecal contents from patients with HTN | fecal contents from patients of normotensive control | oral inoculate | fresh | two times at 1-day interval | 200 µl of the fecal contents/ mice/time | SBP, DBP, HR (tail-cuff blood pressure system) | 10 weeks post-transplantation |
| David J Durgan 1 | 2016 | The United States | Long Evans rats, NA, 8–9 week | 10-day broad spectrum antibiotics treatment | 4 | 4 | cecal contents from hypertensive OSA rats on a high-fat diet | cecal contents from sham donor rats on a high-fat diet | oral inoculate | frozen | 4 consecutive days plus an additional booster inoculation 10 days after the original | 1ml cecal contents/ rat/time | SBP (tail cuff blood pressure system) | between 8 am and noon, before and following 7 and 14 days |

*(Continued)*

**Table 1.** (Continued)

| Author | Year | Country | Animals (species, sex, age) | Antibiotic pretreatment | N (T) | N (C) | Intervention | Control | Route of delivery | Fresh/ Frozen | Times of infusion | Dose of infusion | Outcome measurement | Measurements time |
|---|---|---|---|---|---|---|---|---|---|---|---|---|---|---|
| David J Durgan 2 | 2016 | The United States | Hypertensive OSA rats, NA, 8–9 week | 10-day broad spectrum antibiotics treatment | 4 | 4 | cecal contents from high-fat hypertensive OSA donors | cecal contents from high-fat sham donors | oral inoculate | frozen | 4 consecutive days plus an additional booster inoculation 10 days after the original | 1ml cecal contents/ rat/time | SBP (tail cuff blood pressure system) | between 8 am and noon, before and following 7 and 14 days |
| Piotr Konopelski | 2021 | Poland | WKY rats, Male, 7 week | 5-day antibiotic treatment | 7 | 7 | SHR colon contents | WKY colon contents | intracolonic administration | fresh | twice (days 15 and 16 of the experiment) | 1 ml of colon content/rat/ time | SBP, DBP, HR (telemetry transmitters) | continuous record after the catheter implantation |

WKY, Wistar-Kyoto rats; NA, not applicable; SHRSP, spontaneously hypertensive stroke-prone rats; SBP, systolic blood pressure; SHR, spontaneously hypertensive rats; DBP, diastolic blood pressure; HR, heart rate

**Table 2. SYRCLE's risk of bias tool for each experimental animal studies.**

| | Random Sequence Generation | Baseline Characteristics | Allocation Concealment | Random Housing | Blinding (study Team) | Random Outcome Assessment | Blinding (Outcome Assessors) | Incomplete Outcome Data | Selective Outcome Reporting |
|---|---|---|---|---|---|---|---|---|---|
| | SELECTION BIAS | | | PERFORMANCE BIAS | | DETECTION BIAS | | ATTRITION BIAS | REPORTING BIAS |
| Sareema Adnan | ? | + | - | + | ? | ? | ? | + | + |
| Marta Toral | ? | + | - | + | ? | + | + | + | + |
| Jing Li | ? | + | - | + | ? | + | ? | + | + |
| David J Durgan | ? | + | - | + | ? | ? | ? | + | + |
| Piotr Konopelski | ? | + | - | + | ? | + | + | + | + |

+: low risk of bias -: high risk of bias ?: unclear risk of bias

normotensive controls. One trial was performed in Long Evans rats [18], and the feces were obtained from hypertensive obstructive sleep apnea (OSA) rats on a high-fat diet or high-fat sham rats. One trial was performed in hypertensive OSA rats [18], where the feces were obtained from high-fat hypertensive OSA donors or high-fat sham rats.

## 2.3 Risk of bias assessment

According to the SYRCLE risk of bias tool, the risk of bias was assessed and is displayed in Table 2. No trials were assessed as having a low risk of bias across all domains. For the random sequence generation and blinding domains, all studies were rated as having an unclear risk of bias due to a lack of information. For baseline characteristics and random housing domains, all studies were evaluated as having a low risk of bias. None of the studies described the methods used for allocation concealment, which we assessed as a high risk of bias. Three studies randomly selected animals for outcome assessment [6, 13, 15], but only two studies reported blinding the outcome assessor [13, 15]. All studies were found to sufficiently report complete data, and all expected outcomes were reported. Based on the risk of bias tool, the included studies were rated as high risk.

## 2.4 Quantitative assessment of outcomes

**2.4.1 Systolic blood pressure.** Meta-analysis of SBP was performed in five studies (k = 8) [6, 13–15, 18]. The heterogeneity test showed $I^2$ = 96%, $P$<0.001, thus, the random effect model

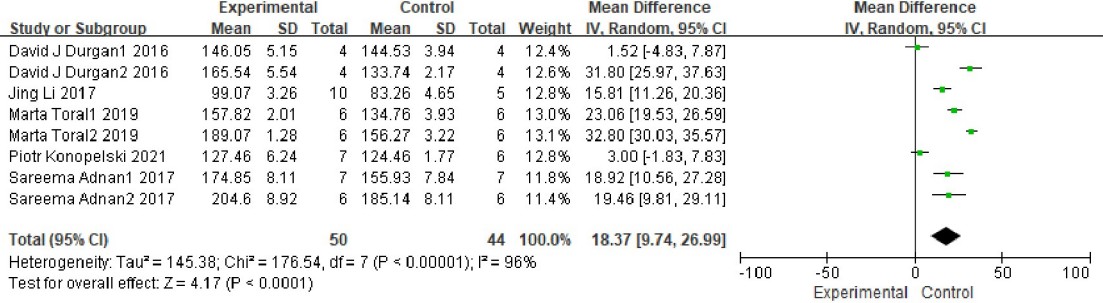

**Fig 2. Forest plot of studies investigating the effect of fecal microbiota transfer on systolic pressure.**

was used for the analysis. The meta-analysis showed that transplantation of fecal bacteria from the hypertensive model can significantly improve the SBP of the host (MD = 18.37, 95%CI: 9.74~26.99, $P$<0.001), presented in Fig 2. Sensitivity analysis was carried out by deleting the studies one by one, and there was no significant change in the comprehensive results, denoting that the results of this study were relatively steady. The funnel plot (Fig 3) was made using the Stata 12.0 software to identify publication bias in the included publications (p for Begg's test = 0.536; p for Egger's test = 0.190), and it showed no potential publication bias for the included studies. Tables 3 and 4 summarized the results of subgroup analysis and sensitivity analysis. Subgroup analyses were performed according to blood pressure measurement, and the tail-cuff blood pressure measurements subgroup showed that transplantation of fecal bacteria from the hypertensive model can cause a significant increase in SBP (MD = 17.47, 95% CI = 7.33–27.62, $P$<0.001; $I^2$ = 92%) [6, 14, 18]. The sensitivity analysis indicated that heterogeneity was significantly reduced (MD = 16.95, 95%CI = 13.26–20.64, $P$<0.001; $I^2$ = 16%, Table 4) after excluding Durgan et al.'s study (k = 2) [18]. Durgan et al.'s study used the rat OSA model, while the others did not, which might have been the source of heterogeneity. The artery direct register subgroup showed similar results (MD = 19.73, 95%CI = 4.19–35.27, $P$ = 0.010; $I^2$ = 98%), but the sensitivity analysis results did not change after excluding studies one by one.

In the frozen subgroup, there was a statistically significant difference between feces from hypertensive donors and feces from normotensive donors (MD = 21.50, 95%CI: 12.69~30.31, $P$<0.001) [13, 14, 18], while there was no statistically significant difference among the fresh feces subgroup (MD = 9.43, 95%CI: -3.12~21.99, $P$ = 0.140, Table 3) [6, 15]. There was substantial statistical heterogeneity in the subgroups, which indicated the type of feces was not the

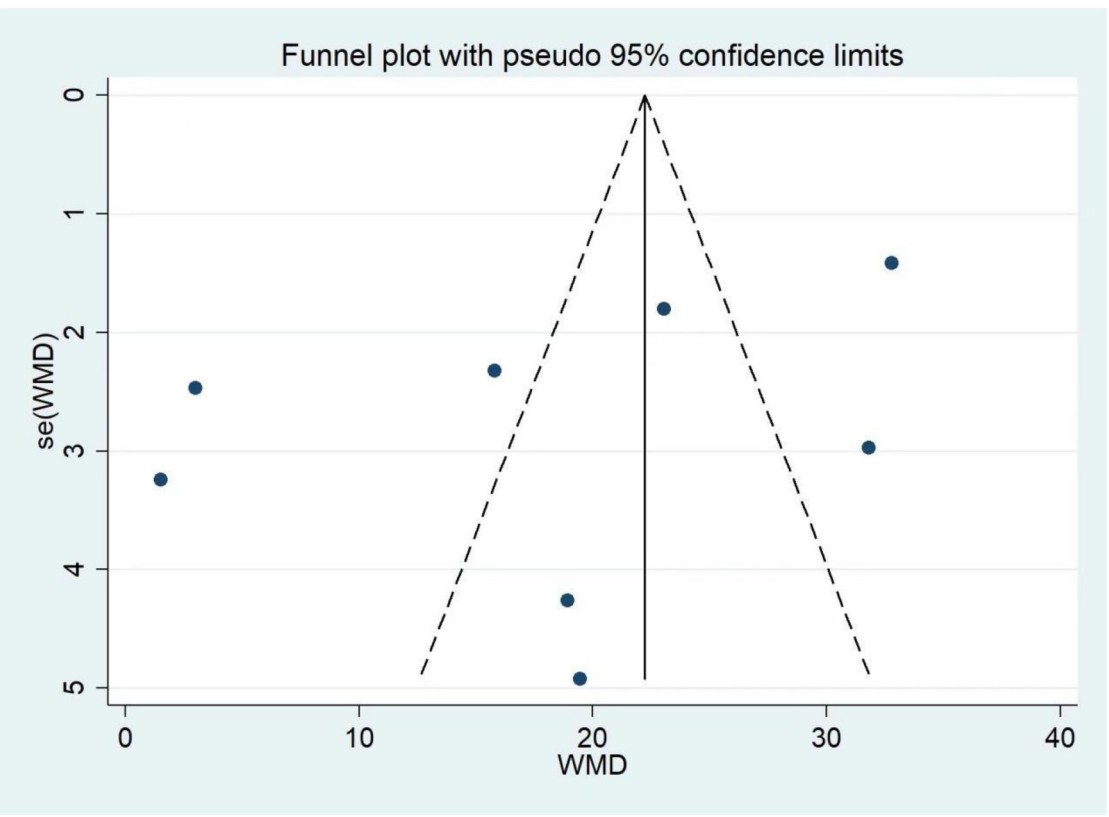

**Fig 3. Publication bias in the studies regarding systolic pressure after fecal microbiota transfer.**

**Table 3. Subgroup analysis of pooled MD for systolic pressure.**

| Categories | No. of trials | No. of animals | Pooled MD (95% CI) | | Heterogeneity | |
|---|---|---|---|---|---|---|
| | | | Random | *P*-value | $I^2$ (%) | *P*-value |
| **BP measurement** | **8** | **94** | **18.37 (9.74, 26.99)** | **<0.001** | **96** | **<0.001** |
| Tail-cuff blood pressure system | 5 | 57 | 17.47 (7.33, 27.62) | <0.001 | 92 | <0.001 |
| Artery direct register | 3 | 37 | 19.73 (4.19, 35.27) | 0.010 | 98 | <0.001 |
| **Type of feces** | **8** | **94** | **18.37 (9.74, 26.99)** | **<0.001** | **58** | **<0.001** |
| Fresh feces | 2 | 28 | 9.43 (-3.12, 21.99) | 0.140 | 93 | <0.001 |
| Frozen feces | 6 | 66 | 21.50 (12.69, 30.31) | <0.001 | 96 | <0.001 |
| **Route of delivery** | **8** | **94** | **18.37 (9.74, 26.99)** | **<0.001** | **96** | **<0.001** |
| Oral gavage | 4 | 50 | 24.29 (16.97, 31.62) | <0.001 | 89 | <0.001 |
| Oral inoculate | 3 | 31 | 16.42 (0.78, 32.05) | 0.040 | 96 | <0.001 |
| Intracolonic administration | 1 | 13 | 0.55 (-1.83, 7.83) | 0.220 | N/A | N/A |
| **Animal model** | **8** | **94** | **18.37 (9.74, 26.99)** | **<0.001** | **96** | **<0.001** |
| Normotensive model | 5 | 62 | 12.48 (3.51, 21.45) | <0.001 | 94 | <0.001 |
| Hypertension model | 3 | 32 | 29.56 (23.55, 35.58) | <0.001 | 71 | 0.030 |
| **Dose of feces** | **8** | **94** | **18.37 (9.74, 26.99)** | **<0.001** | **96** | **<0.001** |
| 1ml fecal contents | 5 | 53 | 18.55 (6.23, 30.87) | 0.003 | 98 | <0.001 |
| 750 µl fecal contents | 2 | 26 | 19.15 (12.84, 25.47) | <0.001 | 0 | 0.93 |
| 250 µl fecal contents | 1 | 15 | 15.81 (11.26, 20.36) | <0.001 | N/A | N/A |

MD, mean difference; CI, confidence interval; BP, blood pressure; N/A, not applicable.

source of heterogeneity. Furthermore, the results of the sensitivity analysis did not change after excluding studies one by one.

In the route of delivery subgroup, the oral gavage subgroup showed high heterogeneity (MD = 24.29, 95%CI: 16.97~31.62, *P*<0.001; $I^2$ = 89%) [13, 14]. After excluding Marta et al.'s study (k = 2) [13], the heterogeneity was distinctly reduced (MD = 16.95, 95%CI = 13.26–20.64, *P*<0.001; $I^2$ = 16%). The antibiotic treatment before transplantation in Marta et al.'s study only lasted for 5 days, while the others demonstrated a longer treatment time, which

**Table 4. Sensitivity analysis on effects of FMT on systolic pressure and diastolic pressure.**

| Outcome | Before sensitivity analysis | | | Remove study | After sensitivity analysis | | |
|---|---|---|---|---|---|---|---|
| | Effect estimate | *P* | $I^2$ (%) | | Effect estimate | *P* | $I^2$ (%) |
| **SBP** | | | | | | | |
| **BP measurement** | | | | | | | |
| Tail-cuff blood pressure system | 17.47 (7.33, 27.62) | <0.001 | 92 | Durgan1 Durgan2 | 16.95 (13.26, 20.64) | <0.001 | 16 |
| **Route of delivery** | | | | | | | |
| Oral gavage | 24.29 (16.97, 31.62) | <0.001 | 89 | Marta1 Marta2 | 19.15 (12.84, 25.47) | <0.001 | 0 |
| **Animal model** | | | | | | | |
| Normotensive model | 12.48 (3.51, 21.45) | <0.001 | 94 | Adnan2 | 32.62 (30.11, 35.12) | <0.001 | 0 |
| **DBP** | 17.65 (12.37, 22.93) | <0.001 | 77 | Marta2 | 15.34 (10.86, 19.83) | 0.090 | 59 |

FMT, fecal bacteria transplantation; SBP, systolic blood pressure; DBP, diastolic blood pressure.

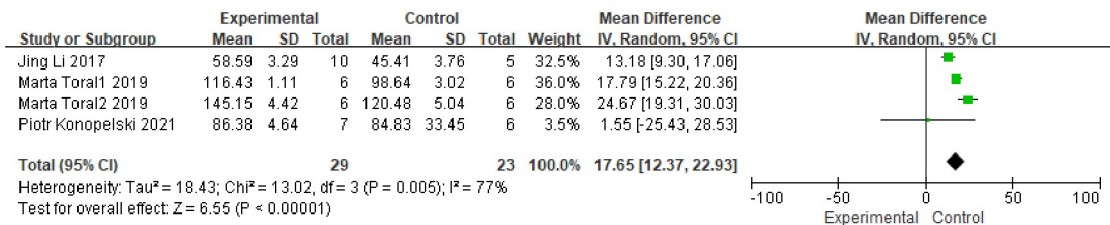

**Fig 4. Forest plot of studies investigating the effect of fecal microbiota transfer on diastolic pressure.**

might have been the source of heterogeneity. The oral inoculated subgroup showed similar results (MD = 16.42, 95%CI = 0.78–32.05, $P<0.001$; $I^2$ = 96%). After excluding studies one by one, the results of the sensitivity analysis did not change. There was only one study in the intra-colonic administration subgroup that did not find a significant difference [15].

In the animal model subgroup, the normotensive model subgroup showed transplantation of fecal bacteria from the hypertensive model caused a significant increase in SBP (MD = 12.48, 95%CI: 3.51~21.45, $P<0.001$) [6, 13–15, 18]. The heterogeneity and results did not change in the sensitivity analysis. The increased SBP in the hypertension model subgroup (MD = 29.56, 95%CI: 23.55~35.58, $P<0.001$) was more pronounced than that in the normotensive model subgroup [13, 14, 18]. After removing Adnan2 et al.'s study [14], the heterogeneity was reduced to 0%, but the statistics did not change. The dose of feces infusion in this study was less than in the other two studies, which might have been the source of heterogeneity.

Then, we performed subgroup analysis according to different dosages, which found that the results weren't influenced by dosage. In the 1ml fecal contents subgroup, transplantation of fecal bacteria from the hypertensive model led to a significant increase in SBP (MD = 18.55, 95%CI = 6.23–30.87, $P$ = 0.003; $I^2$ = 98%) [13, 15, 18], but the sensitivity analysis results did not change after excluding studies one by one. The 750μl fecal contents subgroup showed similar results (MD = 19.15, 95%CI = 12.84–25.47, $P<0.001$; $I^2$ = 0) [14]. The single study in the 250μl fecal contents subgroup also demonstrated comparable results (MD = 15.81, 95% CI = 11.26–20.36, $P<0.001$; $I^2$ = 96%) [6].

**2.4.2 Diastolic blood pressure.**  DBP was performed in three studies (k = 4) [6, 13, 15]. The heterogeneity test showed $I^2$ = 77%, $P<0.001$, therefore, the random effect model was used for the analysis. The meta-analysis gave the result that transplantation of fecal bacteria from the hypertensive model can significantly improve the DBP of the host (MD = 17.65, 95%CI: 12.37~22.93, $P<0.001$), presented in Fig 4. Meanwhile, we noticed that Marta2 et al.'s study [13] was the main resource of heterogeneity, and removing that study led to a marked reduction in heterogeneity ($I^2$ = 59%, $P$ = 0.090). The reason for this may be that the study used the rat hypertension model, while the others did not, which might have been the source of heterogeneity.

**2.4.3 Heart rate.**   HR was performed in three studies (k = 4) [6, 13, 15]. The heterogeneity test showed $I^2$ = 83%, $P<0.001$, therefore, the random effect model was used for the analysis. Meta-analysis showed no significant association between fecal microbiota transfer and HR (MD = 8.74, 95%CI:-16.91~34.38, $P$ = 0.500), presented in Fig 5. Sensitivity analysis as conducted by removing one study after the other indicated that no individual study overly influenced the pooled overall effect in this analysis.

## 3 Discussion

### 3.1 Main findings of this study

This study is the first meta-analysis based on animal experiments to investigate the effect of fecal microbiota transfer on blood pressure. Five studies, covering eight trials, were included

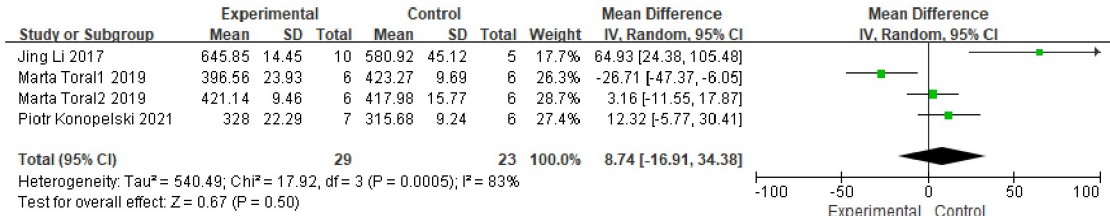

| Study or Subgroup | Experimental | | | Control | | | Weight | Mean Difference IV, Random, 95% CI |
|---|---|---|---|---|---|---|---|---|
| | Mean | SD | Total | Mean | SD | Total | | |
| Jing Li 2017 | 645.85 | 14.45 | 10 | 580.92 | 45.12 | 5 | 17.7% | 64.93 [24.38, 105.48] |
| Marta Toral1 2019 | 396.56 | 23.93 | 6 | 423.27 | 9.69 | 6 | 26.3% | -26.71 [-47.37, -6.05] |
| Marta Toral2 2019 | 421.14 | 9.46 | 6 | 417.98 | 15.77 | 6 | 28.7% | 3.16 [-11.55, 17.87] |
| Piotr Konopelski 2021 | 328 | 22.29 | 7 | 315.68 | 9.24 | 6 | 27.4% | 12.32 [-5.77, 30.41] |
| Total (95% CI) | | | 29 | | | 23 | 100.0% | 8.74 [-16.91, 34.38] |

Heterogeneity: Tau² = 540.49; Chi² = 17.92, df = 3 (P = 0.0005); I² = 83%
Test for overall effect: Z = 0.67 (P = 0.50)

**Fig 5. Forest plot of studies investigating the effect of fecal microbiota transfer on heart rate.**

in this study to systematically evaluate the changes in SBP, DBP, and heart rate in mice/rats after transplantation of hypertensive fecal bacteria. Mice/rats were chosen as research subjects due to the limited number of relevant literature, with four studies using the rats model [13–15, 18], and only one using the mice model [6]. The results demonstrated that transplantation of fecal bacteria from the hypertensive model led to increased SBP in both normotensive and hypertensive animal models, as well as increased DBP. These findings suggest that disruptions in the intestinal flora play a significant role in elevated BP and that modulating the gut microbiota may hold promise as a potential treatment for hypertension.

## 3.2 The relationship between the gut flora and blood pressure

In related studies, increased blood pressure has been associated with an imbalance in gut flora, characterized by a decrease in microbial diversity, an increase in potentially harmful bacteria, and a loss of beneficial bacteria. For example, these studies have shown a significant decrease in the abundance, distribution, and diversity of intestinal flora, an increase in the Firmicutes/Bacteroides ratio (F/B), a decrease in bacteria-producing esters and butyric acid, and a significant increase in bacteria secreting lactic acid [4, 6, 19]. Additionally, Durgan et al. [18] demonstrated that transplantation of cecal contents from hypertensive OSA rats to recipient rats fed a normal chow diet resulted in elevated BP. Li et al. [6] also reported that elevated BP could be transferred through fecal microbiota transfer, as the transfer of fecal microbiota from hypertensive human donors to germ-free mice resulted in increased BP. These findings suggest that the hypertensive phenotype can be transferred through the transfer of caecal contents.

Currently, the transfer of intestinal microflora transfer from hypertensive individuals alters the structure of intestinal microbial flora, primarily leading to increased BP through the following mechanisms. First, fecal microbiota transfer may affect the physiological function of the host through the metabolites SCFAs produced by intestinal microorganisms [20]. SCFAs, such as acetate, propionate, and butyrate, are crucial metabolites for the maintenance of intestinal homeostasis. Notably, butyrate has been shown to protect the heart by preserving gut barrier function, reducing inflammatory response, and inhibiting histone deacetylation [21, 22]. Interestingly, there is a decrease in butyrate-producing bacteria in SHR, and the reduction in butyrate production may disrupt the stability of the intestinal epithelial barrier or enhance vascular tension through the aforementioned mechanisms [4, 23, 24], ultimately resulting in increased BP. Additionally, SCFAs can affect BP by activating olfactory receptors and G protein-coupled receptors when they are absorbed by the intestinal epithelium and enter the host circulatory system. Previous studies have suggested that SCFA receptors participate in the regulation of BP in the kidney [25–27]. Second, inflammatory reactions may serve as another pathological mechanism [28]. Disturbance of intestinal flora can further impair gut barrier function, reduce beneficial bacteria, and stimulate pro-inflammatory cytokines and foam cell formation, thus leading to a cascade of reactions. Inflammation promotes oxidative stress,

which in turn, exacerbates inflammatory reactions. Heightened oxidative stress can result in the oxidation of low-density lipoproteins, which subsequently reduces the release of nitric oxide (NO) from vascular ECs and prostaglandin I-2 (PGI-2), while increasing the levels of vasoconstrictors such as endothelin 1 (ET-1) and thromboxane A2 (TXA2). These changes reduce blood vessel elasticity and compliance, directly contributing to increased BP [29, 30].

### 3.3 Meta-analysis results

The implementation process of FMT has not reached a consensus, and factors such as fecal material type (frozen vs. fresh), and route of delivery may affect the results. To explore the sources of heterogeneity in research results and provide evidence for the clinical implementation of FMT, we conducted subgroup analyses on possible influencing factors. In this meta-analysis, only two studies used fresh fecal, while the majority used frozen fecal bacteria. Frozen feces may offer benefits compared to fresh feces in terms of FMT preparation, storage, monitoring, and delivery [31]. Quantitative polymerase chain reaction analyses have demonstrated that frozen and lyophilized FMT products can be stored for up to 7 months without any alteration in microbial composition or therapeutic efficacy [32]. Previous studies have indicated that frozen fecal transplant is more effective than fresh fecal transplant for ulcerative colitis, due to the reduction of gram-negative bacteria in frozen FMT [33]. In this study, the transplantation of frozen fecal bacteria had a significant impact on the recipient's blood pressure, while the impact of fresh fecal bacteria was not significant. The specific reasons for this are unclear and should be further analyzed in future studies. The route of delivery may also be a key factor affecting the effectiveness of FMT. The delivery methods used in the experiments included in this study are oral gavage, oral inoculate, and intracolonic administration. Only one study used colonic administration, with the results showing that oral administration had a greater impact on blood pressure. Oral administration is the most common route for studies on rodents, as it is relatively simple. Therefore, the impact of different delivery methods on blood pressure may vary for different recipients. The meta-analysis found that the effects were comparable for different dosages of fecal microbiota transplantation, which is similar to the results of Zhang's study [34]. Currently, there is no unified standard for the amount of feces collected. A larger transplantation dose can increase the number and types of donor microbes in the subject's gut while prolonging the stability of probiotics. However, the increased microbial diversity may also produce various metabolic products and damage the intestinal immune barrier, thereby reducing the effectiveness of FMT. Therefore, more well-designed RCTs or stratified cohort studies are still needed to evaluate the impact of different doses.

### 3.4 Systematic review results

In recent years, there has been an increasing interest in using FMT as a potential treatment for cardiovascular diseases, including hypertension. However, the therapeutic benefits of FMT for hypertension are still being explored, and the conclusions from existing studies remain controversial. Among the studies included in this meta-analysis, three trials transplanted fecal bacteria from donors with normal blood pressure into hypertensive mice, but the results were inconsistent. Adnan et al. [13] found that the SBP of SHRSP decreased significantly after fecal transplantation from WKY. Toral et al. [14] found a non-significant decrease in SBP of SHR via FMT, but the difference was not statistically significant. Piotr Konopelski et al. [15] reported that the SBP of both SHR and WKY increased significantly after fecal transplantation into SHR. Due to the limited data available, a meta-analysis was not performed to assess the effectiveness of FMT for hypertension in this study.

### 3.5 Suggestions for future research

The research discovered that fecal microbiota transplantation from hypertensive donors can lead to an increase in blood pressure in the recipient, suggesting a potential transfer of the hypertensive phenotype through the gut contents. This study confirmed a causal relationship between hypertension and the gut microbiota, indicating that FMT may have potential therapeutic implications for hypertension. In the future, precise modulation of the gut microbiota through FMT and supplementation of beneficial gut metabolites may be used for hypertension treatment. However, further research is needed to determine the optimal fecal type and delivery method for human FMT. This may involve studying the effects of different donor types (such as healthy individuals or hypertensive patients) and administration routes (such as oral or colonoscopy) on blood pressure regulation.

### 3.6 Limitations

This meta-analysis has some limitations. Firstly, most of the included studies had a high or unclear risk of bias regarding randomization, allocation concealment, and blinding. The high degree of heterogeneity and low methodological quality weakened the overall quality of the evidence, which could be improved by conducting more randomized prospective trials. Secondly, the search was limited to articles published in English or Chinese, potentially missing relevant reports published in other languages. Third, the small sample size, differences in animal species, and interventions led to high heterogeneity among the studies. Subgroup analyses were performed, but the heterogeneity could not be fully explained. Finally, the research conclusions listed in this review are based on animal experiments, and there is an urgent need for further evidence from human studies.

## 4 Conclusions

This systematic review and meta-analysis provide preliminary data suggesting a relationship between gut microbiota dysbiosis and increased BP. The transplantation of fecal bacteria from hypertensive models can cause a significant increase in systolic and diastolic blood pressure in the host. These findings suggest that addressing gut microbiota dysbiosis may be beneficial for the prevention and treatment of hypertension.

## 5 Materials & methods

### 5.1 Report and register

This study was conducted according to the Preferred Reporting Items for Systematic Reviews and Meta-Analyses (PRISMA) guidelines (S1 Appendix) and registered with PROSPERO (CRD42022353854).

### 5.2 Search strategy

PubMed, EMBASE, Cochrane Library, Web of Science, China National Knowledge Infrastructure (CNKI), WanFang database, Weipu, Embase, and SinoMed were searched without using any filter or limitations to identify all relevant articles on the effects of FMT on BP in animal models. Hand-searching was also conducted. The final search was completed on August 22, 2022. The retrieval strategy combined subject and free words, which were determined by multiple researchers. English search terms include FMT, fecal microbiota transplantation, fecal microbiota transfusion, fecal transplantation, stool microbiota transplantation, stool microbiota transfusion, gut flora, intestinal flora, bacteriotherapy, fecal therapy, fecal bacteriotherapy, intestinal microbiota transplantation, fecal transplant, fecal transfusion, fecal

implantation, fecal implant, fecal instillation, fecal reconstitution; blood pressure, high blood pressure, hypertension, systolic blood pressure, SBP, diastolic blood pressure, and DBP. The retrieval strategy is shown in S1 Table.

### 5.3 Inclusion and exclusion criteria

The study included in the analysis was based on the PICO inclusion criteria. Study design: controlled studies, with no restriction on randomization. Participants: rat or mice, regardless of strain, sex, and modeling method. Intervention: fecal microbiota transfer, with no restrictions on source, dosage, mode of transplantation, or time of intervention. Control: blank control, false stool transplantation, autogenous stool transplantation, or placebo. Outcome: the primary outcome was systolic blood pressure (SBP), and the secondary outcome indicators include diastolic blood pressure (DBP) and heart rate. If multiple time points were reported for an outcome, data from the longest follow-up period were included.

Exclusion criteria were as follows: 1) irrelevant or in vitro experiments; 2) studies combining fecal bacteria with other intervention treatments; 3) repeated reports; 4) studies published in languages other than English or Chinese; 5) observational studies, case reports, protocols, abstracts, conference papers, or reviews; 6) incomplete information or inability to extract data from the literature. If multiple publications reported results from the same research group, only results from the most recent publications were included.

### 5.4 Literature screening and data extraction

Two reviewers independently conducted literature screening and data extraction. All citations were imported into the EndNote X9 software. Titles and abstracts were initially screened, followed by full-text screening according to the inclusion and exclusion criteria. In case of disagreement, a third party was involved to resolve differences. Specific data extracted included the first author, year of publication, country, sample size, animal characteristics (species, age, sex), characteristics of the intervention (source, dose, frequency, transplantation route), outcomes, measurement method, etc. If necessary, the authors of the articles were contacted by email for important information about the study. Since some articles indirectly displayed the data, the software GetData Graph Digitizer (http://getdata-graph-digitizer.com/) was applied to digitize the data and extract sufficient data.

### 5.5 Risk of bias assessment

The risk of bias for each included study was assessed by two reviewers using the Systematic Review Centre for Laboratory animal Experimentation (SYRCLE) risk of bias tool for animal studies [35]. The SYRCLE statement consists of ten items that evaluate whether the studies have low, high, or unclear bias. Studies were categorized as having an overall 'high risk of bias' if at least one key domain was determined to have a high risk of bias, 'unclear' risk of bias if any key areas had an unclear risk, and 'low' risk of bias if all domains were assessed as low-risk [36]. Disagreements in assessments were resolved through consensus or third-party adjudication.

### 5.6 Statistical analyses

Meta-analysis was conducted using the RevMan statistical software 5.4 when there were three or more studies with similar interventions. The unit of analysis for the meta-analysis was the individual experiments (represented as k), with each reference potentially containing multiple independent experiments. As all outcomes were continuous variables, the summary effect

sizes were calculated as means ± standard deviation (SD) and 95% confidence intervals (CI). Heterogeneity between studies was assessed using the $I^2$ statistic. If the test for heterogeneity showed an $I^2$ of <50%, fixed effects models were used, whereas an $I^2$ of ≥50% indicated significant heterogeneity and warranted the use of a random effects model. In cases where the exact number of animals in each group was unclear (e.g., 6–7 animals), attempts were made to contact the authors for clarification. If no response was received, the minimum number was used for the meta-analysis. Subgroup analysis was conducted to explore sources of heterogeneity, considering factors such as types of feces and routes of delivery. Sensitivity analyses were performed by sequentially excluding each study to assess the stability of the results. A significance level of $P<0.05$ was considered statistically significant. If a meta-analysis was not possible, the results would be analyzed and presented descriptively. Egger's test or Begg's test were performed to quantitatively analyze the potential publication bias by STATA version 12.0.

## Supporting information

**S1 Appendix. PRISMA 2020 checklist.**
(DOCX)

**S1 Table. Search strategy for each database.**
(DOCX)

## Author Contributions

**Conceptualization:** Yaqin Chen, Liangwan Chen, Yanjuan Lin.

**Data curation:** Lingyu Lin.

**Funding acquisition:** Liangwan Chen, Yanjuan Lin.

**Investigation:** Lingyu Lin, Shurong Xu, Sailan Li.

**Methodology:** Lingyu Lin, Shurong Xu, Meiling Cai, Sailan Li.

**Project administration:** Liangwan Chen, Yanjuan Lin.

**Writing – original draft:** Lingyu Lin, Shurong Xu, Meiling Cai.

**Writing – review & editing:** Lingyu Lin, Yanjuan Lin.

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
