## [Decision Letter · Decision Letter 0]

3 Sep 2023

PONE-D-23-22042

Effects of fecal microbiota transfer on blood pressure in animal models: A systematic review and meta-analysis

PLOS ONE

Dear Dr. Lin,

Thank you for submitting your manuscript to PLOS ONE. After careful consideration, we feel that it has merit but does not fully meet PLOS ONE’s publication criteria as it currently stands. Therefore, we invite you to submit a revised version of the manuscript that addresses the points raised during the review process.

We look forward to receiving your revised manuscript.

Kind regards,

Li-Ang Lee, M.D., M.Sc.

Academic Editor

PLOS ONE

Additional Editor Comments:

Major Comments: Your manuscript presents a timely systematic review and meta-analysis on the effects of fecal microbiota transfer on blood pressure in animal models. While I commend your thorough literature review and well-structured meta-analysis, there are concerns regarding the low sample sizes and high heterogeneity which could affect the robustness of the meta-analysis results. I recommend that you further enhance the discussion by addressing these limitations and emphasizing potential clinical applications.

Minor Comments:

1. Throughout the manuscript, please refrain from using unnecessary abbreviations.

2. There are several typographical and grammatical errors present, most notably in the abstract. Kindly ensure these are corrected.

3. I suggest adding subtitles in the discussion section. This will provide clarity, particularly when distinguishing between the systematic review and the meta-analysis.

Reviewers' comments:

Reviewer's Responses to Questions

**Comments to the Author**

1. Is the manuscript technically sound, and do the data support the conclusions?

Reviewer #1: Yes

2. Has the statistical analysis been performed appropriately and rigorously? 

Reviewer #1: Yes

3. Have the authors made all data underlying the findings in their manuscript fully available?

Reviewer #1: Yes

4. Is the manuscript presented in an intelligible fashion and written in standard English?

Reviewer #1: Yes

5. Review Comments to the Author

Reviewer #1: Why frozen feces can cause high sbp and fresh feces can’t? and why type of feces was not the source of heterogeneity? The statement here is not reasonable, please explain further on this point.

Please explain more clinical significance for the results of the study? How to treat hypertension by FMT in human in the future?

Which type of feces and which kind of delivery of FMT influence hypertension more? Please explain further.

How to explain recent negative study by Piotr Konopelski, whose conclusion did not comply with your conclusion? Is that related to measurement, FMT delivery, feces type or other kind of confounding factors?

Do the results of the study have practical implications for future studies?

6. PLOS authors have the option to publish the peer review history of their article (what does this mean?). If published, this will include your full peer review and any attached files.

Reviewer #1: **Yes: **Ying Shuo Hsu

---

## [Author Response · Author response to Decision Letter 0]

14 Dec 2023

No. PONE-D-23-22042

Dear Editors and Reviewers:

Thanks for your letter and for the reviewer' comments concerning our manuscript entitled “Effects of fecal microbiota transfer on blood pressure in animal models: A systematic review and meta-analysis” (No. PONE-D-23-22042). Those comments are valuable for revising and improving our paper with important guiding significance. According to the reviewers’ comments, we have made extensive modifications to our manuscript to make our results convincing. In this revised version, changes to our manuscript were marked in yellow highlight. The point-by-point response to the comments is given below. 

Responses to editor-in-chief:

Comment 1: Your manuscript presents a timely systematic review and meta-analysis on the effects of fecal microbiota transfer on blood pressure in animal models. While I commend your thorough literature review and well-structured meta-analysis, there are concerns regarding the low sample sizes and high heterogeneity which could affect the robustness of the meta-analysis results. I recommend that you further enhance the discussion by addressing these limitations and emphasizing potential clinical applications.

Response: Thank you for giving me such valuable advice. In the discussion section, we have included a discussion on the type of feces and route of delivery, as well as supplemented the clinical significance of the study (on Page 11-12 of the revised manuscript). We sincerely hope that this revision will lead to a clearer text.

Comment 2: Throughout the manuscript, please refrain from using unnecessary abbreviations.

Response: Thank you for your reminder. We strongly agree with your suggestions. We thoroughly checked the manuscript and removed unnecessary abbreviations throughout.

Comment 3: There are several typographical and grammatical errors present, most notably in the abstract. Kindly ensure these are corrected.

Response: Thank you for your meticulous work. Your suggestions have proven to be very valuable to us. To improve English and grammar, we have asked native English speakers to revise the manuscript. 

Comment 4: I suggest adding subtitles in the discussion section. This will provide clarity, particularly when distinguishing between the systematic review and the meta-analysis.

Response: We sincerely appreciate the valuable comments. We have added subtitles in the Discussion section. 

Responses to Reviewer #1: 

Comment 1: Why frozen feces can cause high sbp and fresh feces can’t? and why type of feces was not the source of heterogeneity? The statement here is not reasonable, please explain further on this point. 

Response: I am sorry that this part was not clear in the original manuscript. In this study, sub-group analysis was conducted based on the type of feces, and the meta-analysis results showed that in the fresh subgroup, there was no statistically significant difference between feces from hypertensive donors and feces from normotensive donors (MD=9.43, 95%CI: -3.12~21.99, P=0.140; I2=93). In the frozen subgroup, there was a statistically significant difference between feces from hypertensive donors and feces from normotensive donors (MD=21.50, 95%CI: 12.69~30.31, P<0.001; I2=96). However, both subgroups had high heterogeneity, so the type of feces is not the source of heterogeneity. In the results section (on Page 6-7, line 142-145 of the revised manuscript), we have corrected any incorrect statements. Only two trials used fresh fecal transplantation, and the meta-analysis of the combined results of the two groups showed no significant differences. The reason may be related to the small sample size and differences in blood pressure monitoring methods in the two trials. Due to insufficient data, further experimental studies are needed to research the effectiveness of fresh and frozen FMT.

Comment 2: Please explain more clinical significance for the results of the study? How to treat hypertension by FMT in human in the future? 

Response: Joanne et al.'s study summarized the findings on the involvement of the gut microbiota in the host's blood pressure regulation mechanism, and found that gut microbiota imbalance, characterized by a decrease in beneficial bacteria, can affect the host's microbiota-related gene pathways, leading to hypertension. The research discovered that fecal microbiota transplantation from hypertensive donors can lead to an increase in blood pressure in the recipient, suggesting a potential transfer of the hypertensive phenotype through the gut contents. Our study confirmed a causal relationship between hypertension and the gut microbiota, indicating that FMT may have potential therapeutic implications for hypertension. In the future, precise modulation of the gut microbiota through FMT and supplementation of beneficial gut metabolites may be used for hypertension treatment. However, further research is needed to determine the optimal fecal type and delivery method for human FMT. This may involve studying the effects of different donor types (such as healthy individuals or hypertensive patients) and administration routes (such as oral or colonoscopy) on blood pressure regulation. We have added content about the clinical significance of the research in the discussion section on Page 11-12 of the revised manuscript.

References:

1.O'Donnell JA, Zheng T, Meric G, et al. The gut microbiome and hypertension. Nat Rev Nephrol. 2023,19(3):153-167. 

2.Zhong HJ, Zeng HL, Cai YL, et al. Washed Microbiota Transplantation Lowers Blood Pressure in Patients With Hypertension. Front Cell Infect Microbiol. 2021, 11:679624.

Comment 3: Which type of feces and which kind of delivery of FMT influence hypertension more? Please explain further. 

Response: In this systematic review, only two studies used fresh fecal, while the majority used frozen fecal bacteria. Frozen feces may offer benefits compared to fresh feces in terms of FMT preparation, storage, monitoring, and delivery (1). Quantitative polymerase chain reaction analyses have demonstrated that frozen and lyophilized FMT products can be stored for up to 7 months without any alteration in microbial composition or therapeutic efficacy (2). Previous study has indicated that frozen fecal transplant is more effective than fresh fecal transplant for ulcerative colitis, due to the reduction of gram-negative bacteria in frozen FMT (3). In this study, the transplantation of frozen fecal bacteria had a significant impact on the recipient's blood pressure, while the impact of fresh fecal bacteria was not significant. The specific reasons for this are unclear and should be further analyzed in future studies. The route of delivery may also be a key factor affecting the effectiveness of FMT. The delivery methods used in the experiments included in this study are oral gavage, oral inoculate, and intracolonic administration. Oral administration is the most common route for studies on rodents, as it is relatively simple. In our study, only one study used colonic administration, and the meta-analysis showed that oral administration had a greater impact on blood pressure. Therefore, the impact of different delivery methods on blood pressure may vary for different recipients, and targeted research is still needed.

References:

1.Lee CH, Steiner T, Petrof EO, et al. Frozen vs Fresh Fecal Microbiota Transplantation and Clinical Resolution of Diarrhea in Patients With Recurrent Clostridium difficile Infection: A Randomized Clinical Trial. JAMA 2016, 315:142-9. 

2.Jiang ZD, Alexander A, Ke S, et al. Stability and efficacy of frozen and lyophilized fecal microbiota transplant (FMT) product in a mouse model of Clostridium difficile infection (CDI). Anaerobe 2017;48:110-4.

3.Zhu F, Ke Y, Luo Y, Wu J, Wu P, Ma F, Liu Y. Effects of Different Treatment of Fecal Microbiota Transplantation Techniques on Treatment of Ulcerative Colitis in Rats. Front Microbiol. 2021, 12:683234. 

Comment 4: How to explain recent negative study by Piotr Konopelski, whose conclusion did not comply with your conclusion? Is that related to measurement, FMT delivery, feces type or other kind of confounding factors? Do the results of the study have practical implications for future studies?

Response: In the study by Piotr Konopelski et al., it was found that an inter-strain fecal transplant does not affect blood pressure and does not produce long-term changes in gut bacteria composition, which is contrary to the conclusions of our study. We speculate that the reason for this is that in this study, blood pressure in mice was measured using telemetry recordings, fecal transplantation was administered via colonic gavage using fresh fecal material, which differs significantly from the other experiments included in our study. It would be beneficial to replicate this study design in the future and validate the experimental results. Our study results emphasize the potential clinical application of FMT in the management of hypertension, but further research is needed to determine the optimal strategy for implementing FMT in humans. This study confirmed a causal relationship between hypertension and the gut microbiota, indicating that FMT may have potential therapeutic implications for hypertension. In the future, precise modulation of the gut microbiota through FMT and supplementation of beneficial gut metabolites may be used for hypertension treatment. However, further research is needed to determine the optimal fecal type and delivery method for human FMT. This may involve studying the effects of different donor types (such as healthy individuals or hypertensive patients) and administration routes (such as oral or colonoscopy) on blood pressure regulation

We sincerely hope that this manuscript has revised all your comments and suggestions, which we marked in highlight in our changes. We appreciate for editors' and reviewers’ warm work earnestly and hope the correction will meet with approval. We will be happy to edit the manuscript further, based on helpful comments from the reviewers. 

Once again, thank you very much for your comments and suggestions.

Best wishes,

Yanjuan Lin

---

## [Decision Letter · Decision Letter 1]

6 Feb 2024

PONE-D-23-22042R1Effects of fecal microbiota transfer on blood pressure in animal models: A systematic review and meta-analysisPLOS ONE

Dear Dr. Lin,

Thank you for submitting your manuscript to PLOS ONE. After careful consideration, we feel that it has merit but does not fully meet PLOS ONE’s publication criteria as it currently stands. Therefore, we invite you to submit a revised version of the manuscript that addresses the points raised during the review process.

We look forward to receiving your revised manuscript.

Kind regards,

Li-Ang Lee, M.D., M.Sc.

Academic Editor

PLOS ONE

Journal Requirements:

Additional Editor Comments:

Dear Authors,

Following your revisions in response to the initial review, we have thoroughly examined the updated manuscript. It's evident that considerable effort has been made, significantly improving the quality of your work. Nevertheless, we have pinpointed a few minor aspects that still need addressing to fully meet the publication standards. We are optimistic that attending to these details will further elevate the manuscript's caliber.

We appreciate your commitment to enhancing the study and look forward to your continued refinement.

Best regards,

Reviewers' comments:

Reviewer's Responses to Questions

**Comments to the Author**

1. If the authors have adequately addressed your comments raised in a previous round of review and you feel that this manuscript is now acceptable for publication, you may indicate that here to bypass the “Comments to the Author” section, enter your conflict of interest statement in the “Confidential to Editor” section, and submit your "Accept" recommendation.

Reviewer #1: All comments have been addressed

Reviewer #2: All comments have been addressed

2. Is the manuscript technically sound, and do the data support the conclusions?

Reviewer #1: Yes

Reviewer #2: Yes

3. Has the statistical analysis been performed appropriately and rigorously? 

Reviewer #1: Yes

Reviewer #2: Yes

4. Have the authors made all data underlying the findings in their manuscript fully available?

Reviewer #1: Yes

Reviewer #2: Yes

5. Is the manuscript presented in an intelligible fashion and written in standard English?

Reviewer #1: Yes

Reviewer #2: Yes

6. Review Comments to the Author

Reviewer #1: Well written response.

Reviewer #2: This meta-analysis study provides significant insights into the relationship between fecal microbiota transfer and blood pressure regulation in animal models. However, I have several comments that could enhance the clarity and depth of your manuscript.

1. The article presents results from subgroup analyses based on blood pressure measurement methods, fecal material type (frozen vs. fresh), and route of delivery, and animal model. Can the authors elaborate on the rationale behind selecting these specific subgroups for analysis?

2. The visual inspection of funnel plots is a common method to assess publication bias, but it is somewhat subjective. Did you consider complementing this with more objective statistical tests, such as Egger's test or Begg's test, to provide a more robust analysis of publication bias?

3. Have you ever thought that the dosage of fecal infusion might influence the outcome? Although Zhang et al.’s study in 2020 (https://doi.org/10.1007/s13238-019-00684-8) ever reported that the fecal weight was not well correlated with the amount of enriched microbiota, what results caused by dosage of fecal infusion do you think your research will bring?

7. PLOS authors have the option to publish the peer review history of their article (what does this mean?). If published, this will include your full peer review and any attached files.

Reviewer #1: **Yes: **Ying-Shuo Hsu

Reviewer #2: No

---

## [Author Response · Author response to Decision Letter 1]

20 Feb 2024

No. PONE-D-23-22042R1

Dear Editors and Reviewers:

Thanks for your letter and for the reviewer' comments concerning our manuscript entitled “Effects of fecal microbiota transfer on blood pressure in animal models: A systematic review and meta-analysis” (No.PONE-D-23-22042R1). Those comments are all valuable and very helpful for revising and improving our paper, as well as the important guiding significance to our researches. We have studied comments carefully and have marked in the yellow highlight in the paper. The point-by-point response to the comments is given below. 

Responses to Reviewer #2:

Comment 1: The article presents results from subgroup analyses based on blood pressure measurement methods, fecal material type (frozen vs. fresh), and route of delivery, and animal model. Can the authors elaborate on the rationale behind selecting these specific subgroups for analysis?

Response: I am sorry that this part was not clear in the original manuscript. The implementation process of FMT has not reached a consensus, and factors such as fecal material type (frozen vs. fresh), and route of delivery may affect the results. To explore the sources of heterogeneity in research results and provide evidence for the clinical implementation of FMT, this study conducted subgroup analyses on possible influencing factors.

Studies have shown that the storage conditions of fecal samples can affect the composition of the microbiota. European consensus conference on faecal microbiota transplantation in clinical practice indicates that fresh feces within 6 hours should be used; before further preparation, feces should be stored in sealed containers at 2-8°C (Quality of evidence: moderate. Strength of recommendation: strong). Preparation of frozen feces (Quality of evidence: moderate. Strength of recommendation: strong). Frozen transplant materials can simplify the microbiota transplantation process without reducing efficacy and safety. Our study found that the use of frozen fecal microbiota transplantation had a greater impact on blood pressure. There are many routes for fecal microbiota transplantation, mainly divided into oral and nasal (nasogastric or capsule ingestion) or colonic implantation. The oral and nasal route is widely used and easy to implement. Colonic implantation is consistent with the recipient's physiology, but there is an issue that fecal liquids have a short retention time in the gastrointestinal tract. Currently, there is no clear evidence to support which transplantation route is most appropriate. In our manuscript, there were more studies using the oral route than the colonic implantation, and the results showed that the oral route had a greater impact on recipient blood pressure.

The blood pressure measurement methods included tail-cuff blood pressure system and carotid artery direct register. Carotid artery direct register is directly transmitted arterial pressure waveforms through sensors, while tail-cuff blood pressure system measures blood flow within arteries. Differences in measurement methods could lead to differences in results. Therefore, we conducted subgroup analyses based on blood pressure measurement. The studies included in our manuscript involved subjects including both normal blood pressure and hypertensive subjects. We also conducted subgroup analyses based on animal models (normotensive model and hypertension model) to explore the effects of fecal microbiota transplantation on subjects with normal blood pressure and hypertensive subjects, respectively. 

In the discussion section, we have briefly supplemented the reasons for conducting subgroup analysis (on Page 10-11 of the revised manuscript). We sincerely hope that this revision will lead to a clearer text.

References:

1.European consensus conference on faecal microbiota transplantation in clinical practice[J].Gut Journal of the British Society of Gastroenterology, 2017.

2.Cammarota,Giovanni,Ianiro,et al.Faecal microbiota transplantation in clinical practice[J].Gut Journal of the British Society of Gastroenterology, 2018.

Comment 2: The visual inspection of funnel plots is a common method to assess publication bias, but it is somewhat subjective. Did you consider complementing this with more objective statistical tests, such as Egger's test or Begg's test, to provide a more robust analysis of publication bias?

Response: Thank you for your reminder. We strongly agree with your suggestions. We used the Egger’s and Begg’s test of STATA version 12.0 to assess the possibility of publication bias. Egger’s and Begg’s test shows that p value was higher than 0.05 (p for Begg’s test = 0.536; p for Egger’s test = 0.190), indicating that there was no significant publication bias observed in the selected studies (Figure 3). In the results section (on Page 6, line 127-129 of the revised manuscript), we have rephrased the publication bias results. 

Comment 3: Have you ever thought that the dosage of fecal infusion might influence the outcome? Although Zhang et al.’s study in 2020 (https://doi.org/10.1007/s13238-019-00684-8) ever reported that the fecal weight was not well correlated with the amount of enriched microbiota, what results caused by dosage of fecal infusion do you think your research will bring?

Response: We agree with your viewpoint that the dosage of fecal microbiota transplantation would affect the results. We conducted a subgroup analysis based on the dosage of fecal microbiota transplantation, and the results showed that transplantation of fecal bacteria from the hypertensive model led a significant increase in SBP (MD=18.55, 95%CI=6.23-30.87, P=0.003; I2=98%) in the 1ml fecal contents subgroup, but the sensitivity analysis results did not change after excluding studies one by one. The 750µl fecal contents subgroup showed similar results (MD=19.15, 95%CI=12.84-25.47, P<0.001; I2=0). The single study in the 250µl fecal contents subgroup also demonstrated comparable results (MD=15.81, 95%CI=11.26-20.36, P<0.001; I2=96%). In the results section (on Page 7-8, line 164-171 of the revised manuscript), we have rephrased the publication bias results. 

In the discussion section, we have discussed the impact of fecal microbiota dosage on the results (on Page 11 of the revised manuscript). The amount of feces does not have a proportional relationship with the number of microbes, and there are large individual differences. Currently, there is no unified standard for the amount of feces collected. A larger transplantation dose can increase the number and types of donor microbes in the subject's gut while prolonging the stability of probiotics. However, the increased microbial diversity may also produce various metabolic products and damage the intestinal immune barrier, thereby reducing the effectiveness of FMT. Therefore, more well-designed RCTs or stratified cohort studies are still needed to evaluate the impact of different doses. Considering the total transplantation dose may also affect the results of FMT, the combined design of dose and frequency can also be included in future research objectives. 

In the manuscript, we have removed the 8th reference because it is not covered in SCI-indexed journals and it does not affect the overall content of the article (Fang H, Fu L, Wang J. Protocol for Fecal Microbiota Transplantation in Inflammatory Bowel Disease: A Systematic Review and Meta-Analysis. Biomed Res Int. 2018:8941340.). 

We sincerely hope that this manuscript has revised all your comments and suggestions, which we marked in highlight in our changes. We appreciate for editors' and reviewers’ warm work earnestly and hope the correction will meet with approval. We will be happy to edit the manuscript further, based on helpful comments from the reviewers. 

Once again, thank you very much for your comments and suggestions.

Best wishes,

Yanjuan Lin

---

## [Decision Letter · Decision Letter 2]

6 Mar 2024

Effects of fecal microbiota transfer on blood pressure in animal models: A systematic review and meta-analysis

PONE-D-23-22042R2

Dear Dr. Lin,

We’re pleased to inform you that your manuscript has been judged scientifically suitable for publication and will be formally accepted for publication once it meets all outstanding technical requirements.

Kind regards,

Li-Ang Lee, M.D., M.Sc., Ph.D., F,I.C.S.

Academic Editor

PLOS ONE

Additional Editor Comments (optional):

Dear Authors,

I am writing to inform you about the progress of your manuscript, "Effects of Fecal Microbiota Transfer on Blood Pressure in Animal Models: A Systematic Review and Meta-Analysis" (PONE-D-23-22042R2). After a thorough review process involving myself and a dedicated team of reviewers, I am pleased to announce that your manuscript has been favorably evaluated.

All concerns previously identified have been satisfactorily addressed, leading us to the unanimous decision that your work holds significant value and is a suitable candidate for publication in our esteemed journal. It is with great pleasure that I recommend moving forward with the publication process.

I would like to take this opportunity to thank you for your patience and diligence in addressing the feedback provided. Your commitment to excellence has been evident throughout this process.

Warmest regards,

Reviewers' comments:

Reviewer's Responses to Questions

**Comments to the Author**

1. If the authors have adequately addressed your comments raised in a previous round of review and you feel that this manuscript is now acceptable for publication, you may indicate that here to bypass the “Comments to the Author” section, enter your conflict of interest statement in the “Confidential to Editor” section, and submit your "Accept" recommendation.

Reviewer #2: All comments have been addressed

2. Is the manuscript technically sound, and do the data support the conclusions?

Reviewer #2: Yes

3. Has the statistical analysis been performed appropriately and rigorously? 

Reviewer #2: Yes

4. Have the authors made all data underlying the findings in their manuscript fully available?

Reviewer #2: Yes

5. Is the manuscript presented in an intelligible fashion and written in standard English?

Reviewer #2: Yes

6. Review Comments to the Author

Reviewer #2: The authors have responded point by point to reviewers' comments. The revised manuscript is more complete and scientifically sound. I think this version of the manuscript can be accepted and published in this journal.

7. PLOS authors have the option to publish the peer review history of their article (what does this mean?). If published, this will include your full peer review and any attached files.

Reviewer #2: No

---

## [Editor Report · Acceptance letter]

12 Mar 2024

PONE-D-23-22042R2 

PLOS ONE

Dear Dr. Lin, 

I'm pleased to inform you that your manuscript has been deemed suitable for publication in PLOS ONE. Congratulations! Your manuscript is now being handed over to our production team.

Kind regards, 

on behalf of

Dr. Li-Ang Lee 

Academic Editor

PLOS ONE